# Functional Characterization of the *CpNAC1* Promoter and Gene from *Chimonanthus praecox* in *Arabidopsis*

**DOI:** 10.3390/ijms24010542

**Published:** 2022-12-29

**Authors:** Xiaoyan Zhao, Jiahui Zhao, Qing Yang, Min Huang, Yangjing Song, Mingyang Li, Shunzhao Sui, Daofeng Liu

**Affiliations:** Chongqing Engineering Research Center for Floriculture, Key Laboratory of Horticulture Science for Southern Mountainous Regions of Ministry of Education, College of Horticulture and Landscape Architecture, Southwest University, Chongqing 400715, China

**Keywords:** NAC family genes, transcription factor, wintersweet, promoter, abiotic stress

## Abstract

The NAC (NAM, ATAF, and CUC) gene family is one of the largest plant-specific transcription factor families. Its members have various biological functions that play important roles in regulating plant growth and development and in responding to biotic and abiotic stresses. However, their functions in woody plants are not fully understood. In this study, we isolated an NAC family member, the *CpNAC1* promoter and gene, from wintersweet. CpNAC1 was localized to the nucleus and showed transcriptional activation activity. qRT-PCR analyses revealed that the gene was expressed in almost all tissues tested, with the highest levels found in mature leaves and flower buds. Moreover, its expression was induced by various abiotic stresses and ABA treatment. Its expression patterns were further confirmed in *CpNAC1pro:GUS* (*β-glucuronidase*) plants. Among all the transgenic lines, *CpNAC1pro*-D2 showed high GUS histochemical staining and activity in different tissues of *Arabidopsis*. Furthermore, its GUS activity significantly increased in response to various abiotic stresses and ABA treatment. This may be related to the stress-related *cis*-elements, such as ABRE and MYB, which clustered in the *CpNAC1pro*-D2 segment, suggesting that *CpNAC1pro*-D2 is the core segment that responds to abiotic stresses and ABA. In addition, *CpNAC1-*overexpressed *Arabidopsis* plants had weaker osmosis tolerance than the wild-type plants, demonstrating that *CpNAC1* may negatively regulate the drought stress response in transgenic *Arabidopsis*. Our results provide a foundation for further analyses of NAC family genes in wintersweet, and they broaden our knowledge of the roles that NAC family genes may play in woody plants.

## 1. Introduction

During their long period of growth and development, plants are inevitably affected by various adverse environmental factors, including abiotic stresses, such as cold, high temperature, drought, and high levels of salt, which are important limiting factors [1]. To resist adversity, plants can respond by transmitting stress signals, which involves complex physiological and genetic changes, including changes in the expressions of a series of stress-resistance-related genes. These are precisely controlled by molecular regulation networks in plants to ensure growth, development, and survival [2,3,4]. Transcriptional regulation is the most important part of gene expression, and it is a key step in the plant response to stress [5]. Transcription factors (TFs) are a class of protein factors that play a role in this process by combining with specific *cis*-acting elements to regulate the expressions of downstream genes. As key regulators, TFs are widely involved in the stress response, helping to regulate the expression of stress-resistance-related genes in signal transduction pathways [6]. The TF families involved in responses to abiotic stress in plants include MYB, bZIP, DREB, NAC, and WRKY [7].

NAC proteins constitute one of the largest families of plant-specific TFs. They share a conserved N-terminal DNA-binding domain called the NAC domain, whose name is derived from three TFs (no apical meristem (NAM), *Arabidopsis thaliana* transcription activation factor (ATAF1/2), and cup-shaped cotyledon proteins (CUC2)) [8]. In 1996, the NAC TF was first cloned from *Petunia hybrida*, and it was found that it plays a role in the formation and differentiation of the apical meristem [9]. Since then, NAC TFs have been identified in many plants [10]. The structure of NAC TFs is characterized by the highly conserved NAC domain at the N-terminus, which is responsible for binding to DNA and other proteins. The C-terminal parts of NAC proteins are highly diverse and do not contain any known protein domains [11]. The NAC domain consists of approximately 150–160 amino acid residues and five subdomains (A–E). The A, C, and D subdomains are highly conserved, and the nuclear localization signals are located in the C and D subdomains, which mainly bind to DNA and determine the characteristics of NAC TFs. The B and E subdomains are not strongly conserved, which is related to the functional diversity of NAC TFs [12,13]. 

The NAC family members are widely involved in a variety of biological functions in plants, such as growth [14,15], fiber development [16], secondary wall synthesis [17,18], lateral root development [19], embryonic development [20], hormone responses [21], cell expansion [22,23], leaf senescence [24,25,26], pathogen resistance [27,28], and fruit ripening [29]. They also play key roles in the regulation of abiotic stresses, such as cold, high temperature, drought, and high levels of salt [30]. For example, *GmNAC8* overexpressed lines exhibit higher drought tolerance than wild-type plants, whereas *GmNAC8* knockout lines are drought-sensitive, indicating that *GmNAC8* acts as a positive regulator of drought tolerance in soybean [31]. The ectopic expression of *HaNAC1* from *Haloxylon ammodendron* promotes growth and drought tolerance in transgenic *Arabidopsis* [32]. *SlNAP1* positively regulates salt tolerance in tomato by regulating ion homeostasis and ROS metabolism [33]. Moreover, the overexpression of *TaNAC2L* in transgenic *Arabidopsis* may improve heat tolerance by regulating the expression of stress-responsive genes [34]. *SlNAC1* can be induced by cold, heat, high levels of salt, osmotic stress, and other conditions in tomato, and its overexpression can enhance resistance to cold stress in transgenic *Arabidopsis* [35]. Unlike the above NACs, *ATAF1* in *Arabidopsis* is a negative regulator of defense responses against necrotrophic fungal and bacterial pathogens [36]. *OsNAC2* overexpressed lines of rice have lower resistances to high levels of salt and drought [37]. The knockout mutants of *ATAF1* show a higher tolerance to heat stress than the wild type, revealing a negative regulatory role [38].

To date, NAC TFs have been widely studied in model plants and major crops, such as *Arabidopsis thaliana* [11], rice [39], wheat [40], *Zea mays* [41], soybean [42], pepper (*Capsicum annuum*) [43], and peanut (*Arachis hypogaea*) [44]. However, studies on NAC family genes in woody plants are still limited. Wintersweet (*Chimonanthus praecox*) is a unique, precious, traditional ornamental landscape plant in China. It blooms in cold winters, and it shows strong cold and drought resistance [45]. Some genes related to the abiotic stress responses [46,47], flower development [48,49], senescence [40,50], shoot branching [51,52], and volatile compound regulation [48,53] of wintersweet have been identified and characterized. In this study, we isolated an NAC family member, the *CpNAC1* promoter and gene, from wintersweet [54], and we identified its tissue-specific and induction expression patterns. Moreover, its function was analyzed in overexpression transgenic lines of *Arabidopsis*.

## 2. Results

### 2.1. Isolation and Characterization of CpNAC1

*CpNAC1* was screened using the wintersweet transcriptome database [54], and full-length cDNA sequences of *CpNAC1* were successfully cloned. *CpNAC1* has an open reading frame (ORF) of 891 bp, which encodes a peptide of 296 amino acid residues. The molecular weight and theoretical isoelectric point of *CpNAC1* were 35 kDa and 8.54, respectively. Multiple sequence alignments indicated that CpNAC1 showed high similarity with its homologous sequences, including ZmNAC1 (*Zea mays*), GmNAC2 (*Glycine max*), ANAC072 (*Arabidopsis thaliana*), ATAF1 (*Arabidopsis thaliana*), ATAF2 (*Arabidopsis thaliana*), GmNAC4 (*Glycine max*), and NnNAC2 (*Nelumbo nucifera*). CpNAC1 has a conserved N-terminal domain of approximately 160 amino acid residues with five subdomains (A–E), and the C-terminal domain is highly differentiated (Figure 1A). The phylogenetic tree of CpNAC1 and other NAC proteins was constructed using MEGA 7.0. The results classified NAC proteins into three groups: SNAC, VND, and CUC. CpNAC1 was clustered with SNAC subgroup proteins, and it was most closely related to GmNAC2 and ATAF1 (Figure 1B).

### 2.2. CpNAC1 Is a Nuclear Protein with Transcriptional Activation Activity in Yeast

To identify the location of CpNAC1, the control vector *35S:GFP* and recombinant vector *35S:CpNAC1-GFP* were transformed into tobacco epidermal cells using a needleless syringe. In a confocal microscopic analysis, GFP fluorescence was observed only in the nucleus of *35S*:*CpNAC1-GFP*, while it was observed in both the cytoplasm and nucleus in the controls (Figure 2A). In addition, a yeast assay was used to investigate the transcriptional activity of CpNAC1. The plasmids pGBKT7*-CpNAC1*, pGBKT7*-VP*, and pGBKT7 were introduced into the AH109 yeast strain. The transformed yeast strain grew normally on SD/-Trp medium, indicating that these plasmids were introduced successfully. When grown on SD/-His+X-α-gal selective medium, yeast cells harboring pGBKT7*-CpNAC1* and pGBKT7*-VP* grew and showed α-galactosidase activity, whereas those harboring pGBKT7 did not (Figure 2B). Collectively, these results indicate that CpNAC1 is a nuclear-localized protein with transcriptional activation abilities in yeast. 

### 2.3. Expression Patterns of CpNAC1 in Wintersweet

qRT-PCR was performed to analyze the expression patterns of *CpNAC1* in the different tissues and flower development stages of wintersweet. As shown in Figure 3A, the expression of *CpNAC1* was most abundant in mature leaves, followed by cotyledons, pistils, and young leaves. Its expression in different flower stages was also analyzed. It was expressed during flower development in all stages, with the highest expression in flower buds, at which point it was nearly 4-fold higher than in other stages. 

### 2.4. Isolation and cis-Element Analysis of the CpNAC1 Promoter

We isolated approximately 1.4 kb genomic DNA sequences upstream of the *CpNAC1* gene, including 87 bp of the 5′ untranslated region of mRNA and the 1242 bp promoter sequence, *CpNAC1pro* (Figure 4A). The analysis of the *cis*-acting regulatory elements showed that *CpNAC1pro* contained basic promoter elements, such as TATA-box and CAAT-box; the abscisic acid (ABA) response element abscisic-acid-responsive element (ABRE); the jasmonic acid (JA) response element TGACG-motif; the growth hormone response element AuxRR-core; the gibberellin (GA) response element GARE-motif; and the P-box, MYB, and MYC motifs, which respond to dehydration ABA signals and abiotic stress signals (Figure 4; Appendix A). To further verify the function of *CpNAC1pro*, the promoter sequence was divided into three segments: *CpNAC1pro-*D1*/*D2*/*D3, with 1025 bp, 513 bp, and 266 bp, respectively (Figure 4B).

### 2.5. Analysis of CpNAC1 Promoter Activity

To verify the promoter activity and tissue-specific expression pattern of *CpNAC1*, *CpNAC1pro* and the three segments of *CpNAC1pro*-D1/D2/D3 were fused to the *GUS* reporter gene in order to generate transgenic *Arabidopsis* plants. A histochemical GUS staining of these plants showed that all transgenic lines had GUS activity in leaves and roots after seed germination from 1 to 10 days (Figure 5A). Among them, *CpNAC1pro* had weaker activity than *CpNAC1pro*-D1/D2/D3, while *CpNAC1pro*-D2 had the strongest activity. In addition, the GUS activity of the different tissues was observed in the transgenic lines (Figure 5B). In roots, the activity of *CpNAC1pro*-D2 was approximately twice that of *CpNAC1pro*-D1 and *CpNAC1pro*-D3 (Figure 5C). In stems, it was approximately twice as high as that of *CpNAC1pro* (Figure 5C). In leaves, it was approximately three times higher than that of *CpNAC1pro* and *CpNAC1pro*-D3 (Figure 5C). In petals, it was approximately four times higher than that of *CpNAC1pro* (Figure 5C). The negative control wild-type *Arabidopsis* had almost no GUS activity in all tissues; the positive control pCAMBIA1305.1:*GUS* transgenic *Arabidopsis* had relatively low GUS activity in all tissues (Figure 5C). In general, these results indicate that GUS activity was the highest in the *CpNAC1pro*-D2 transgenic lines. 

### 2.6. Expression Profiles of CpNAC1 under Abiotic Stress and Hormone Treatments

To investigate the effects of abiotic stresses and exogenous hormones on *CpNAC1* expression, the transcriptional levels of *CpNAC1* under various abiotic stresses, including cold (4 °C), high-temperature (42 °C), salt (NaCl 1 M), drought (30% PEG6000), and ABA (50 µM) treatments, which were chosen according to the *cis*-acting elements of the *CpNAC1* promoter, were detected using qRT-PCR (Figure 6).

For the high-temperature treatment, the expression of *CpNAC1* was slightly upregulated at the 0.25 h time point; it then decreased at 1 h. For the cold treatment, the expression of *CpNAC1* was gradually induced and reached a peak at 12 h, which was about six times higher than that before the treatment. For the salt and ABA treatments, the expression level of *CpNAC1* was induced and peaked at 1 h, and then it decreased to the lowest point at 12 h but remained higher than the expression level before the treatment. For the PEG6000 treatment, the expression level of *CpNAC1* was induced at 1 h and reached a peak at 6 h, with a 4-fold higher expression than that before the treatment.

To further study the expression patterns of the *CpNAC1* promoter under abiotic stresses and hormone treatments, the responses of *CpNAC1pro* and *CpNAC1pro-*D1/D2/D3 to the above treatments were also examined (Figure 7). Ten-day-old homozygous *CpNAC1pro:GUS* transgenic *Arabidopsis* seedlings were used for the assay. The expression of the *GUS* gene was significantly increased when the transgenic seedlings were treated with PEG, ABA, and NaCl (Figure 7A). The GUS activity of *CpNAC1pro* was significantly increased in the ABA, NaCl, and PEG treatments (Figure 7B–D). However, the activity of *CpNAC1pro-*D2 was significantly increased in all treatments (Figure 7B–F). This may be related to the stress-related *cis*-elements, such as ABRE and MYB, which clustered in the *CpNAC1pro-*D2 segment (Figure 4B). These results suggest that *CpNAC1pro-*D2 is the core segment that responds to abiotic stresses and ABA. 

### 2.7. Heterologous Overexpression of CpNAC1 Negatively Regulates Tolerance to Osmotic Stress in Transgenic Arabidopsis 

To further investigate the function of *CpNAC1*, it was overexpressed in *Arabidopsis,* and transgenic lines were obtained via hygromycin selection and PCR identification. Three homozygous T3 transgenic lines (OE8, OE9, and OE7) with high, medium, and low *CpNAC1* expression levels were selected for further phenotypic analyses and PEG osmotic stress treatment (Figure 8A); wild-type plants were used as controls. Wild-type and the three transgenic *Arabidopsis* lines were treated with 30% PEG6000. After 3 days of PEG treatment, both the wild-type and transgenic *Arabidopsis* were partially wilted and dehydrated (Figure 8B). The proline and chlorophyll contents of the transgenic lines were significantly lower than those of the wild-type line (Figure 8C,D). The relative electrical conductivities were 42.51%, 55.66%, 59.03%, and 53.3% in the wild-type and OE (OE8, OE9, and OE7) lines, respectively (Figure 8E). These results demonstrate that *CpNAC1* may negatively regulate the osmotic stress response in transgenic *Arabidopsis*.

## 3. Discussion

NAC TFs, unique to plants, are one of the largest groups of TFs, and they play an important role in plant responses to biotic and abiotic stresses [55]. Therefore, the study of the molecular mechanisms underlying the responses of NAC genes to various stresses will aid the development of plants with increased stress resistance in the future. Here, we report the isolation of an NAC family gene, CpNAC1, from wintersweet. It shows a high similarity to sequences of other plant SNAC subfamily genes, and it has a high number of conserved domains (Figure 1A). Phylogenetically, it clusters with GmNAC2 and ATAF1, belonging to the evolutionary clade SNAC (Figure 1B). *GmNAC2* functions as a negative regulator during abiotic stress, and it participates in ROS signaling pathways [56]. *ATAF1* in *Arabidopsis* as a transcriptional regulator negatively regulates the defense response against pathogens [36], thermorecovery [38], and the expression of stress-responsive genes under drought stress in *Arabidopsis* [57]. 

Like other NAC TFs, such as CaNAC064 [58], ZmNAC55 [59], and GhNAC2 [60], the CpNAC1-GFP fusion protein is located in the nucleus (Figure 2A), which concords with its putative function as a transcriptional regulator. In addition, CpNAC1 shows transcriptional activation in yeast (Figure 2B). Moreover, *CpNAC1* is expressed specifically in different tissues and stages, with the highest expressions in mature leaves and sprouts (Figure 3). The results suggest that *CpNAC1* may function in flower organ development and abiotic stress in early plant growth. In maize, the expression of *ZmSNAC1* is significantly higher in roots and leaves in the seedling and early flowering stages than in other organs [61]. The expression of *CaNAC064* of pepper (*Capsicum annuum* L.) is abundant in roots but low in fruits and seeds [58]. *CpNAC68* has different levels of expression, being the highest in old leaves during the bloom stage [45]. In sum, the expression levels of NAC TFs in different plant organs and developmental stages are different. This may be related to their functional diversity. 

Studies have shown that 20–25% of NAC genes respond to at least one or several stresses [39]. *DgNAC1*-overexpressed chrysanthemum is obviously more resistant to salt than the wild type [62]. The overexpression of *HhNAC54* of *Hibiscus hamabo* in *Arabidopsis thaliana* significantly increases its tolerance to salt [63]. In our study, the expression of *CpNAC1* was significantly induced by the cold, NaCl, PEG, and ABA treatments (Figure 6), suggesting that the gene may be involved in resistance to these stressors. Further, the *cis*-element analysis of the *CpNAC1* promoter showed that it contained 11 ABRE, 2 MYB, and 4 MYC elements (Appendix A), which respond to dehydration ABA signals and abiotic stress signals [64]. We hypothesized that *CpNAC1* is involved in abiotic stress tolerance and is related to the ABA pathway. The MYB binding site (MBS) element in the *ZmSOPro* region of maize is responsible for ABA- and drought-stress-induced expressions [65]. In the GUS histochemical staining analysis, we found that almost the whole seedling was stained from day 1 to 10 days after germination, and GUS activity was observed in nearly all organs, including flowers, in transgenic *Arabidopsis* (Figure 5), indicating that this gene may be constitutively expressed during both vegetative and reproductive growth [65]. These results are inconsistent with the specific expression results of *CpNAC1* in the tissues of wintersweet. *CpNAC1* is derived from the genome of wintersweet. There are differences in organ structure between woody plants and herbaceous plants, and the upstream regulatory sequences of the two plants are different, so the efficiency and location of the promoter may also be different. In addition, GUS activity in *CpNAC1pro* was significantly increased after the ABA, NaCl, and PEG treatments (Figure 7). However, the activity in *CpNAC1pro-*D2 was significantly increased under all treatments, which may be related to the stress-related *cis*-elements, such as ABRE and MYB (Figure 4 and Figure 7). *CpNAC1pro-*D2 was the core segment responding to abiotic stresses and ABA. The promoters of multiple stress-responsive genes have a number of regulatory elements that respond to multiple stresses, such as ABRE, MYB, and MYC; these are the potential targets for regulating the stress-inducible expression of transgenes in transgenic plants [66]. This is basically consistent with our observations of the expression of *CpNAC1* under different stress treatments (Figure 6), but different results were obtained for the cold treatment, which may be related to the absence of cold response elements in the promoter of *CpNAC1*. Moreover, the expression of the *CpNAC1* promoter and gene were significantly induced by the PEG and ABA treatments, and the overexpression of the *CpNAC1* gene decreased tolerance to osmotic stress in *Arabidopsis*. Studies have also shown that transgenic plants may recover the phenotype if osmotic stress duration is prolonged, which is negatively correlated with the concentration and treatment time of the PEG solution [19]. We hypothesize that *CpNAC1* might negatively regulate osmotic stress tolerance by regulating ABA-related genes. Previous studies have shown that *OsNAC095* and *OsNAC2* play negative roles in drought stress tolerance in transgenic rice [37,67]. Hence, our study lays a good foundation for a further analysis of NAC family genes in wintersweet. Additionally, the identification and functional characterization of the *CpNAC1* promoter and gene broaden our knowledge of how it responds to abiotic stresses.

## 4. Materials and Methods

### 4.1. Plant Materials and Growth Conditions

Wintersweet (*Chimonanthus praecox* [L.] Link) in the 4–6 leaf stage was used in this study. The plants were grown in a nursery at Southwest University, Chongqing, China. Tissues (roots, stems, cotyledons, young leaves, mature leaves, inner petals, outer petals, stamen, and pistils) and flowers at different stages (sprout, flower bud, display petal, bloom initiation, bloom, and wither periods [68]) were collected, and the samples were immediately frozen in liquid nitrogen, followed by storage at −80 °C until RNA isolation.

Wild-type *Arabidopsis* (ecotype Columbia−0) was used for plant transformation. *Arabidopsis* seeds were sterilized, rinsed in sterile distilled water, sown on solid Murashige–Skoog medium (MS), vernalized at 4 °C for 3 days, and then grown for 12 days in an environment under a 16 h light/8 h dark photoperiod at a temperature of 22 ± 1 °C and 70% relative humidity. After 10 days, the seedlings were planted in a mixture of vermiculite and peat (1:1 ratio) and maintained in a growth chamber (22 °C, 16/8 h day/night photoperiod).

### 4.2. Cloning of the CpNAC1 Gene and Promoter of Wintersweet

Total RNA was extracted using a Plant RNAprep Pure kit (Tiangen Biotech, Beijing, China), and it was used to synthesize the first-strand cDNA with a PrimerScript RT reagent kit with gDNA Eraser (TaKaRa, Dalian, China). PCR amplification was carried out using cDNA as the template and *CpNAC1-F/R* (Appendix A) as specific primers with high-fidelity polymerase (SMARTer RACE 5′ 3′ Kit). Then, the samples were sent to Huada Gene Technology Co., Ltd. (Shanghai, China) for sequencing.

Total DNA was extracted from the leaves using the CTAB method [69]. The 5′-upstream region of *CpNAC1* was isolated using Universal Genome Walker™ 2.0 (TaKaRa). The specific primers *CpNAC1pro-F/R* (Appendix A) were designed based on the sequence at the 5′ end of *CpNAC1*. Then, the cloned segment was inserted into a pMD19-T vector for validity sequencing. Using the PlantCARE program (http://bioinformatics.psb.ugent.be/webtools/plantcare/html/, accessed on 18 August 2022) found in plant *cis*-acting regulatory DNA elements, we analyzed and annotated the *cis*-elements in the sequence of the *CpNAC1* promoter.

### 4.3. Subcellular Localization and Transactivation Activity Analysis

The ORFs of *CpNAC1* without stop codons were cloned into the pCAMBIA1300*-GFP* vector harboring CaMV35S using the *BamH*I and *Sac*I sites to generate the *35S*:*CpNAC1*-*GFP*. The specific primers used are listed in Appendix A. The fusion vector *35S*:*CpNAC1*-*GFP* or empty vector *35S:GFP* (control) was transformed into epidermal cells from tobacco (*Nicotiana benthamiana*) in the 4–6 leaf stage, using the *Agrobacterium tumefaciens* strain GV3101. They were incubated for 36 h under a dark and moist environment, and GFP fluorescence was observed under a laser scanning confocal microscope [70].

To perform the transactivation activity in yeast cells, the CDS sequence of the *CpNAC1* gene was constructed onto the pGBKT7 vector using *Xma*I and *Sal*I sites to form a plasmid pGBKT7-*CpNAC1.* The plasmid pGBKT7*-VP* and empty vector pGBKT7 were used as positive and negative controls, respectively. The plasmids were introduced into the AH109 yeast strain using the LiAc-mediated transformation method according to the manufacturer’s instructions (Clontech). Positive transformants were selected on SD/-Trp plates and then cultured on SD/-His and SD/-His/X-α-gal plates to assess transcriptional activity; this was performed according to the yeast protocol handbook (Clontech).

### 4.4. Expression Patterns of CpNAC1 in Wintersweet

Wintersweet seedlings in the 4–6 leaf stage were used to study the *CpNAC1* expression patterns under different abiotic and hormone treatments [45]. Cold and heat treatments were conducted by transferring the seedings to a growth chamber set at 4 °C or 42 °C under a 16 h light/8 h dark cycle. For the salt, drought, and ABA stress treatments, the seedlings were grown in soil irrigated with 1 M NaCl, 30% PEG6000, or 50 µM ABA [47]. Then, we collected the leaves at 0 h, 0.25 h, 1 h, 6 h, and 12 h after treatment. Control plants were mock-treated at room temperature (25 °C) and with water. All samples were immediately frozen in liquid nitrogen and stored at −80 °C until RNA isolation. The expression patterns of *CpNAC1* in the collected samples were determined using quantitative real-time (qRT)-PCR on a Bio-Rad CFX96 Real-time system using the *qCpNAC1-F/R* specific primers (Appendix A), according to the methods described by Liu et al. [49]. Wintersweet *Actin* and *Tublin* genes served as internal references. For each tested tissue, three biological replicates were included by harvesting samples from three different plants.

### 4.5. GUS Histochemical and GUS Activity Assays

The CaMV35S promoter of the pCAMBIA1305.1 vector was replaced by the *CpNAC1* promoter using *Pst*I and *Nco*I sites to form a plasmid *CpNAC1pro:GUS.* To conduct a promoter deletion analysis, we constructed a series of vectors using promoter fragments of different lengths (D1, D2, and D3) to replace the CaMV35S promoter; these were named *CpNAC1pro-*D1/D2/D3*:GUS.* To further analyze the expression patterns of the *CpNAC1* promoter, the recombinant plasmid was cloned into the *Agrobacterium tumefaciens* strain GV3101 and transformed into *Arabidopsis* using the floral-dip method [71]. The seeds of the transgenic plants were selected on solid MS containing 50 μg/L hygromycin, and then they were confirmed using real-time PCR. The wild-type and pCAMBIA1305.1:*GUS* transgenic *Arabidopsis* were used as negative and positive controls.

To explore the expression patterns of the *GUS* gene driven by the *CpNAC1* promoter and each deletion fragment, we analyzed the different growth stages (seeds 1 to 10 days after germination) and tissues (inflorescences, flowers, petals, calyx, stamens, styles, stems, fruit pods, rosette leaves, and stem leaves) from the transgenic *Arabidopsis* plants using GUS histochemical staining [72]. The samples were immersed in GUS staining solution and incubated overnight at 37 °C. After staining, the plants were completely depleted of chlorophyll using 70% ethanol and then photographed using a stereomicroscope (Nikon, Japan). GUS enzyme activity was detected following a previously reported method [73].

Ten-day-old transgenic *Arabidopsis* seedlings were transferred into solid MS containing 50 µM ABA, 150 mM NaCl, and 10% PEG6000 for 24 h to analyze GUS enzyme activity. For the cold and heat treatments, the 10-day-old transgenic *Arabidopsis* seedlings were moved to a growth chamber set at 4 °C or 42 °C for 24 h or 4 h, respectively. The expression levels of *GUS* in the *CpNAC1pro:GUS* transgenic *Arabidopsis* under hormonal and abiotic stresses were determined using quantitative real-time (qRT)-PCR on a Bio-Rad CFX96 real-time system using *GUS*-*F/R* specific primers (Appendix A). The *Arabidopsis Actin* gene served as an internal reference. For each tested tissue, three biological replicates were included by harvesting samples from three different plants, and qRT-PCR was performed using three technical replicates for each sample.

### 4.6. Osmotic Stress of Transgenic Arabidopsis

The plant overexpression vector was constructed by inserting *CpNAC1* into the pCAMBIA1300 vector (containing the CaMV35S promoter) using specific primers pCAMBIA1300*-CpNAC1-F/R* (Appendix A) with *Sac*I and *Xbal* sites and then transformed into the *Agrobacterium tumefaciens* strain GV3101 via electroporation. *Arabidopsis* was transformed as described in Section 4.5. The seeds of the wild-type and *CpNAC1* transgenic *Arabidopsis* were sterilized and sown on MS plates; after 14 days, they were transferred to a mixture of vermiculite and peat (1:1 ratio) and placed in a greenhouse under a 16 h light/8 h dark cycle at 22 °C for 3 weeks. The wild-type and three transgenic *Arabidopsis* lines were irrigated with 30% PEG6000 for 3 days. The phenotype was observed and photographed. The proline content was calculated as described in a previous study [74]. The chlorophyll content of the plants was determined according to the method devised by Mokhtar et al. [75]. The relative electrical conductivity was determined according to previously published methods [76,77]. The experiment was repeated three times, each time with three control pots and nine treatment pots with four seedlings per pot.

### 4.7. Statistical Analysis

Statistical analyses were performed using SPSS software, and the significance of differences between the control and transgenic *Arabidopsis* plants was analyzed using Student’s *t*-test. A *p* value of 0.05 was considered to be significant. Statistical significance was tested using Duncan’s test at 0.05 probability levels.

## Figures and Tables

**Figure 1 ijms-24-00542-f001:**
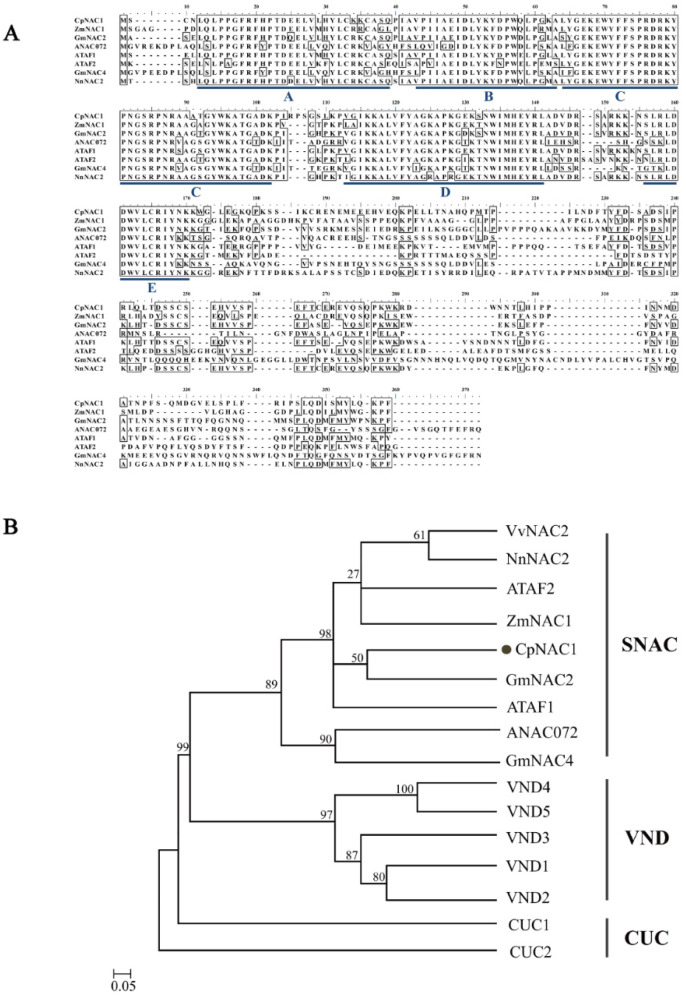
Sequence analysis of CpNAC1. (**A**) Multiple sequence alignment of CpNAC1 and its orthologs from other plants. The N-termini of CpNAC1 conserved domains in five subdomains (A–E) are marked with blue lines below the sequence alignment. (**B**) Phylogenetic tree of CpNAC1 and NAC proteins. The phylogenetic tree was constructed using MEGA 7.0 software and the neighbor-joining (NJ) method (1000 bootstraps). CpNAC1 is indicated by a dot. *Zea mays* ZmNAC1 (ABY67929.1), *Glycine max* GmNAC2 (NP_001240958), GmNAC4 (NP_001238424), ANAC072 (OAO97067), *Arabidopsis thaliana* ATAF1 (CAA52771), ATAF2 (AAM65967), VND1 (OAP07379.1), VND2 (OAO96839.1), VND3 (OAO91397.1), VND4 (OAP17154.1), VND5 (OAP18936.1), CUC1 (ACB31158.1), CUC2 (ACB31182.1), *Nelumbo nucifera* NnNAC2 (XP_010268987), *Chimonanthus praecox* CpNAC1, and *Vitis vinifera* VvNAC2 (XP_002285870.2).

**Figure 2 ijms-24-00542-f002:**
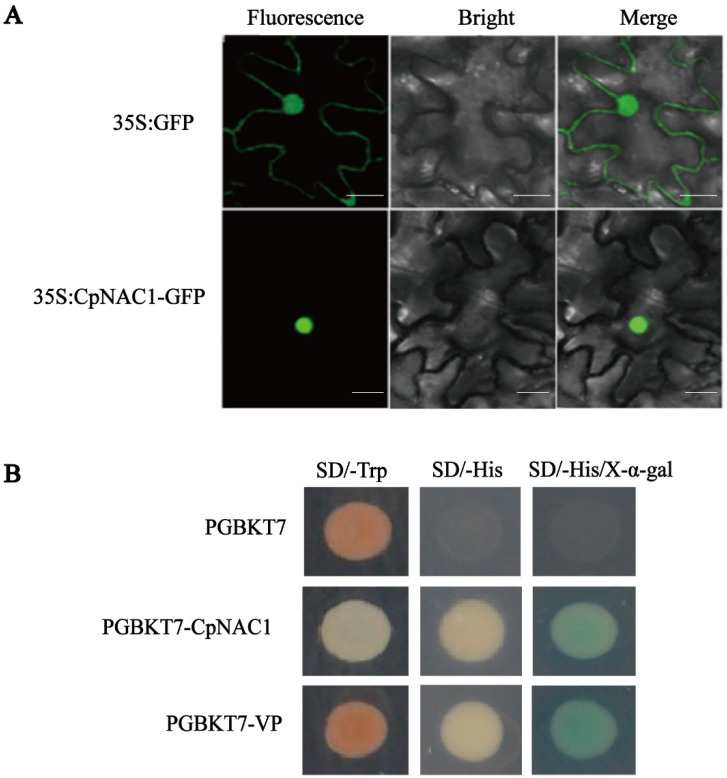
Subcellular localization and transactivation activity analysis of CpNAC1. (**A**) Subcellular localization of CpNAC1 protein. The fusion proteins *35S:GFP* and *35S:CpNAC1*-*GFP* were expressed in tobacco leaf epidermis cells via *Agrobacterium*-mediated infection. *35S:GFP* was used as the control. Bars denote 50 µm. (**B**) Transactivation activity analysis of CpNAC1. The plasmids of pGBKT7-*CpNAC1*, pGBKT7-*VP* (positive controls), and pGBKT7 (negative control) were transformed into the AH109 yeast strain. Growth and β-galactosidase activity were examined on SD/-Trp, SD/-His, and SD/-His/X-α-gal media.

**Figure 3 ijms-24-00542-f003:**
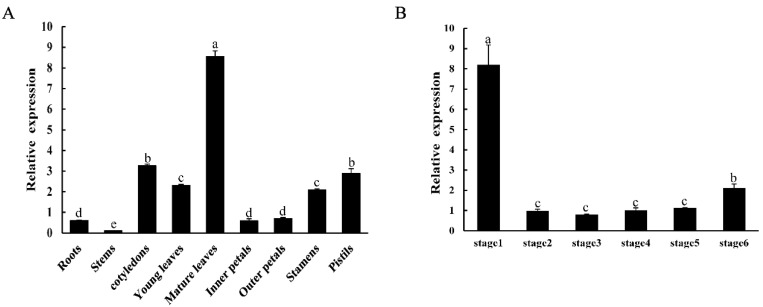
Expression patterns of *CpNAC1* in wintersweet. (**A**) Expressions of *CpNAC1* in different tissues, including roots, stems, cotyledons, young leaves, mature leaves, inner petals, outer petals, stamen, and pistils. (**B**) Expressions of *CpNAC1* in different flower development stages (stages 1–6): stage 1, sprout period; stage 2, flower-bud period; stage 3, display-petal period; stage 4, bloom-initiation period; stage 5, bloom period; stage 6, wither period (Sui et al., 2012). *CpActin* and *CpTublin* were used as internal controls. Data represent means of three biological repeats ± standard deviations (SDs). Different lowercase letters indicate significant differences (*p* < 0.05).

**Figure 4 ijms-24-00542-f004:**
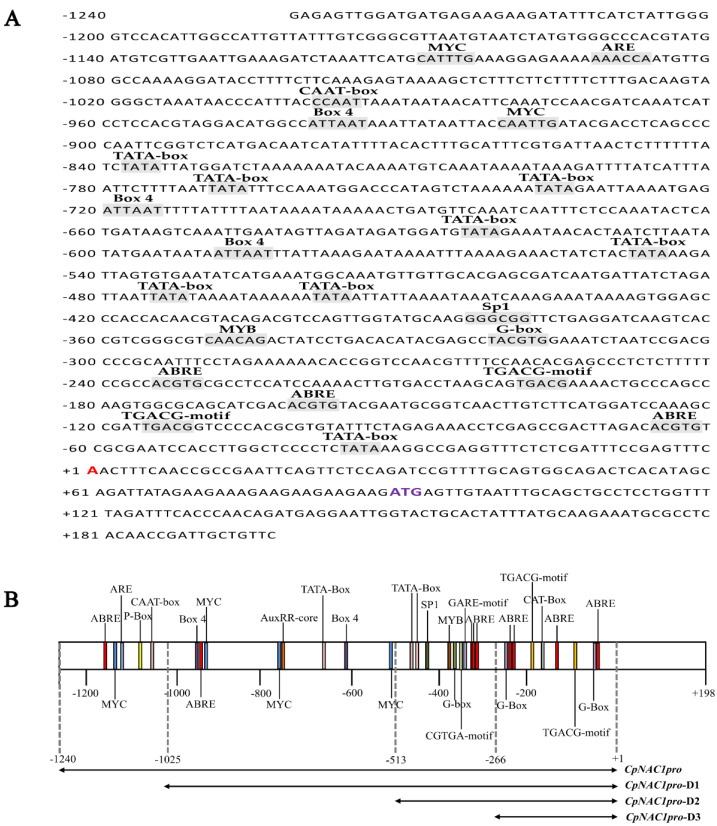
Isolated genomic DNA sequences and different segments with various *cis*-elements of *CpNAC1* promoter. (**A**) Schematic diagram of the *CpNAC1* promoter with putative *cis*-element sites. (**B**) Schematic diagram of *CpNAC1* promoter divided into different segments. The dotted line indicates the position of the deletion segment: *CpNAC1pro-*D1, *CpNAC1pro-*D2, and *CpNAC1 pro-*D3.

**Figure 5 ijms-24-00542-f005:**
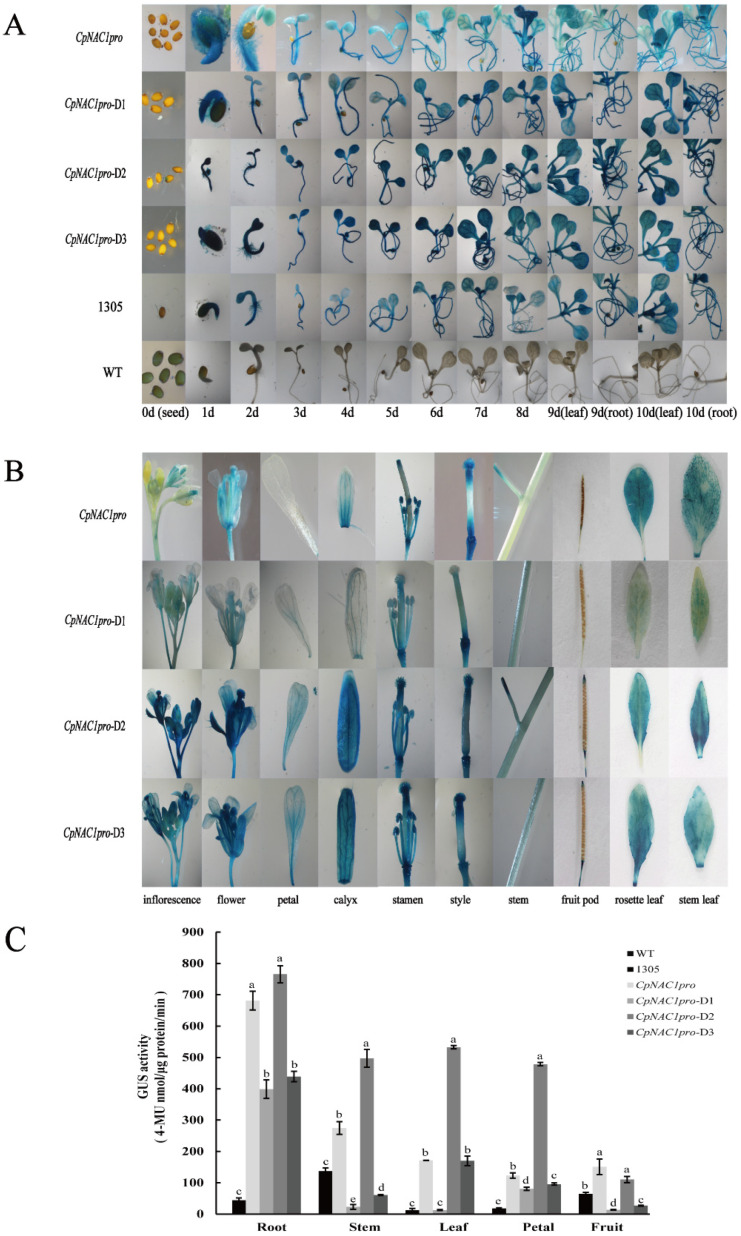
GUS activity of the different segments of the *CpNAC1* promoter in *Arabidopsis.* (**A**) GUS activity under the control of *CpNAC1pro* or *CpNAC1pro*-D1/D2/D3 in transgenic *Arabidopsis* seeds from 1 to 10 days after germination. T3 homozygous lines were used for GUS staining analysis. (**B**) Tissue specificity of GUS histochemical staining for transgenic *Arabidopsis*. (**C**) GUS activity of different tissues in different transgenic *Arabidopsis* lines. Data represent means of three biological repeats ± standard deviations (SDs). Different lowercase letters indicate significant differences (*p* < 0.05).

**Figure 6 ijms-24-00542-f006:**
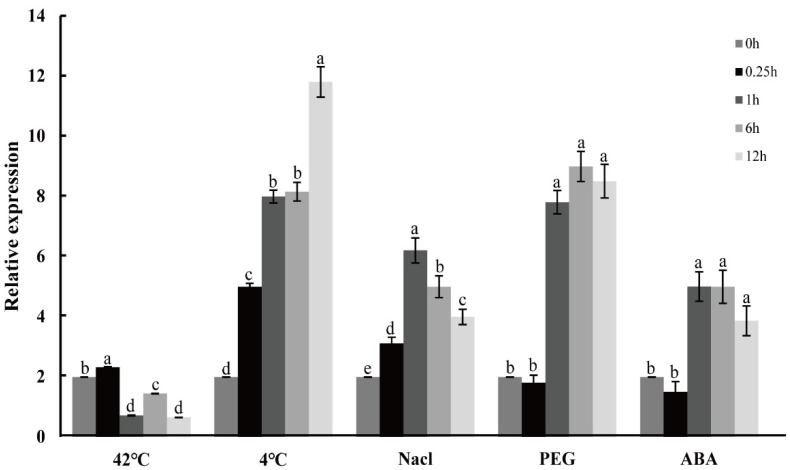
Expression patterns of *CpNAC1* in response to abiotic stress. The wintersweet plants were exposed to treatments at 42 °C, 4 °C, 1 M NaCl, 30% PEG6000, and 50 µM ABA. Data represent means of three biological repeats ± standard deviations (SDs). Different lowercase letters indicate significant differences (*p* < 0.05).

**Figure 7 ijms-24-00542-f007:**
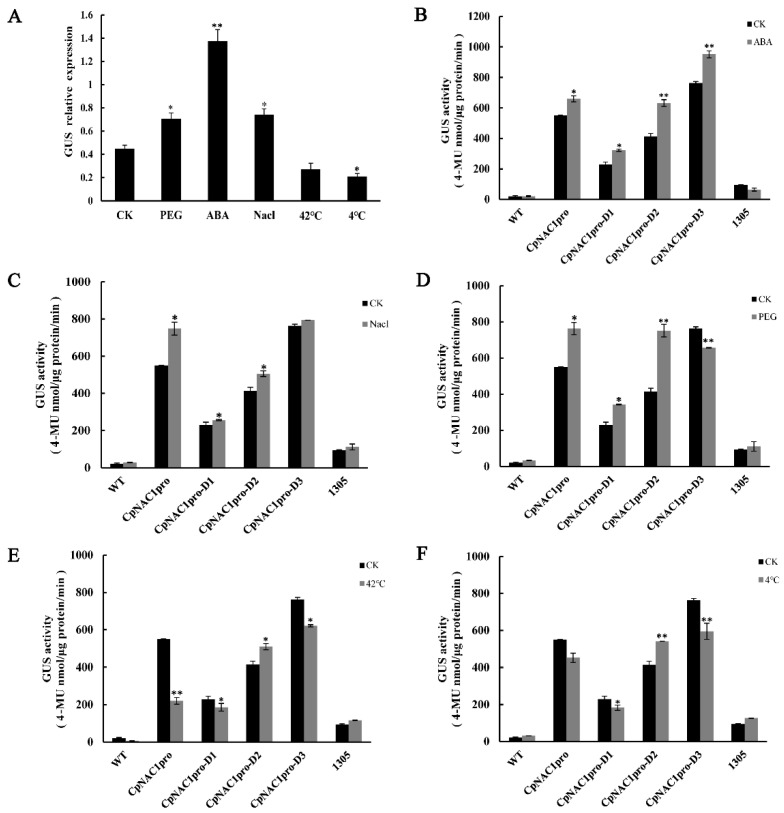
*CpNAC1* promoter activity in response to abiotic stress and hormone treatments. (**A**) Expression patterns of the *GUS* gene in *CpNAC1pro:GUS* transgenic *Arabidopsis* plants. (**B**–**F**) Transgenic *Arabidopsis* GUS enzyme activity after different treatments. Ten-day-old T3 homozygous transgenic *Arabidopsis* seedlings were treated with 50 µM ABA, 150 mM NaCl, 10% PEG for 24 h; heat stress at 42 °C for 6 h; or cold stress at 4 °C for 24 h. Plants grown under normal conditions were used as controls. Data represent means of three biological repeats ± standard deviations (SDs). Asterisks denote statistically significant differences compared to controls, * *p* < 0.05, ** *p* < 0.01.

**Figure 8 ijms-24-00542-f008:**
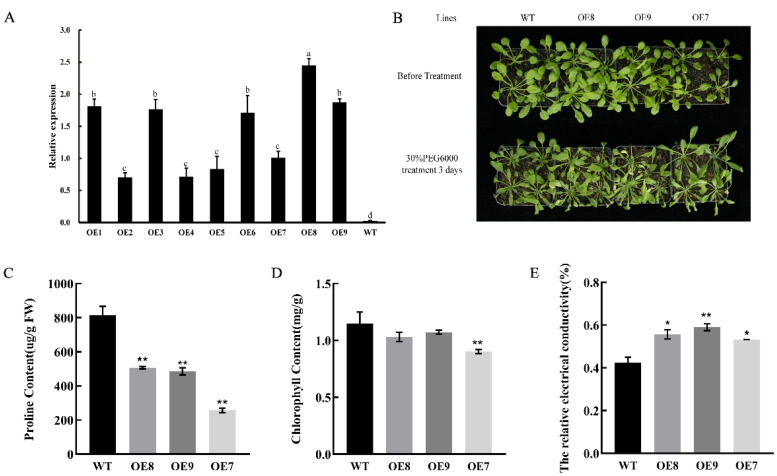
Heterologous overexpression of *CpNAC1* negatively regulates tolerance to osmotic stress in transgenic *Arabidopsis*. (**A**) Relative expression of *CpNAC1* in wild-type and transgenic *Arabidopsis*. Data represent means of three biological repeats ± standard deviations (SDs). Different lowercase letters indicate significant differences (*p* < 0.05). (**B**) Phenotypes of wild-type and *CpNAC1* transgenic *Arabidopsis* under 30% PEG6000 treatment. (**C**–**E**) Proline content, chlorophyll content, and relative electrical conductivity of wild-type and *CpNAC1* transgenic *Arabidopsis* under 30% PEG6000 treatment. Data represent means of three biological repeats ± standard deviations (SDs). Asterisks denote statistically significant differences compared to controls, * *p* < 0.05, ** *p* < 0.01.

## Data Availability

Not applicable.

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
