# Peer review of "Functional Characterization of the CpNAC1 Promoter and Gene from Chimonanthus praecox in Arabidopsis"

_ijms, 2022, doi:10.3390/ijms24010542_

Round 1

Reviewer 1 Report

In the present study the authors showed that the CpSNAC1 gene, a NAC family from wintersweet, played an important role in regulating the drought stress response in Arabidopsis. For that, gene expression, promoter activity and gene overexpressing and other techniques were used. I consider that the authors' findings will contribute to enhance our understanding on the role of the NAC family in the responds to abiotic stresses in wood plants, and that this paper has a merit to be published in IJMS. However, the manuscript needs to be corrected the points described below.

1) Results. p5, L134-135: the control vector 35S:GFP and  recombinant vector 35S S::CpSNAC1-GFP were . Here, the colons need to be unified. There are also similar problems in other parts of the full text.

2) Fig 2A. p5 L147-148: Scale bars were omitted.

3) Fig 2B. p5 L147-148: Result of transformed yeast strains on the SD/-Trp medium needs to be presented as the control in the figure 2 and the corresponding result analysis.

4) Methods of difference significance analysis of each result need to be indicated in Figure legends or Methods.

5) Fig 8 p12 L288: The expression detection of 3 transgenic lines needs to be provided.

Author Response

Response to Reviewer 1 Comments

Point 1: Results. p5, L134-135: the control vector 35S:GFP and recombinant vector 35S S::CpSNAC1-GFP were …. Here, the colons need to be unified. There are also similar problems in other parts of the full text. 

Response 1: Thank you for your guidance, we have changed it in line 128-129 and other parts of our manuscript, as “35S:GFP, 35S:CpNAC1-GFP

Point 2: Fig 2A. p5 L147-148: Scale bars were omitted. 

Response 2: Thank you for your guidance, we have added scale bars in Fig 2A and improved it in Figure legends in line 145, as “Bars denote 50 µm”.

Point 3: Fig 2B. p5 L147-148: Result of transformed yeast strains on the SD/-Trp medium needs to be presented as the control in the figure 2 and the corresponding result analysis.

Response 3: Thank you for your guidance, we have improved Fig 2B in line 141 and added corresponding result analysis in line 134 of our manuscript, as “The transformed yeast strain grew normally on SD/-Trp medium, indicating that these plasmids were introduced successfully”.

Point 4: Methods of difference significance analysis of each result need to be indicated in Figure legends or Methods.

Response 4: Thank you for your guidance, we have added methods of difference significance analysis of each result in each Figure legends and materials and methods in line 461.

Point 5: Fig 8 p12 L288: The expression detection of 3 transgenic lines needs to be provided.

Response 5: Thank you for your guidance, we have added the expression detection of 3 transgenic lines Fig 8A in line 273 and improved it in Figure legends in line 276, as “Relative expression of CpNAC1 in wild-type and transgenic Arabidopsis”.

Reviewer 2 Report

Zhao et al. Characterized the CpSNAC1 transcription factor promoter and gene from Chimonanthus praecox in Arabidopsis in relation with abiotic stresses including ABA treatment. This work has scientific merit but need to fix some points  before publication in any scientific journal.

Major points:

-          Lack of background/rationale behind the choice of the gene of interest. There is no discussion how many  homologs of NAC TF are present in Chimonanthus praecox and their biological relevance in C. praecox.

-          Figure legends less descriptive specially Figure 2 legend. In General Figure legends need to be explained more to clear the message, specially figure 2 part B, BP and VP are not defined…

-          Expression of CpSNAC gene could be first presented and discussion about other isoforms as transcriptomic data available and then follow by isolation and characterization….

-          It is not clear why author name CpSNAC instead of CpNAC… S trends for what!!

-          Do the author try to prepare transgenic line in endogenous species (C. praecox)? What is the biological relevance to study CpSNAC…. Authors claim it would be helpful to know the mechanism of abiotic stress tolerance that is far from what author presented in this study.

-          Phenotypic experiments are not sufficient to conclude that CpNAC negatively regulates the osmotic tolerance… more experiment can be done i.e., root growth assay in vitro control conditions and stress conditions… the data presented is not clear enough to see the phenotype… what the rationale between choosing stress conditions for example concentration of ABA and Nacl…

-          What will happen if the osmotic stress prolonged and is it possible to recover the phenotype?

-          How yeast cells were transformed .. method is not described and  also lack of information on the method used to study the transcriptional activity assay in yeast.

-          Which method used for RT-qPCR analysis. The results of qPCR is not well discussed in discussion that deserved to be discussed more clearly in discussion.

-          What kind of statistical test used to test the significant level that need to be mentioned in figure legends or in methodology.

Finally I would suggest a slight modification of  the title: Functional characterization of the CpSNAC13 promoter and gene from Chimonanthus praecox in Arabidopsis

Minor points:

-          Space between word and figure citation (line 250, 260, 282…..). Several others!

-          Line 356 ONAC should be OsNAC

-          Moderate English check needed

Author Response

Response to Reviewer 2 Comments

Point 1: Lack of background/rationale behind the choice of the gene of interest. There is no discussion how many  homologs of NAC TF are present in Chimonanthus praecox and their biological relevance in C. praecox

Response 1: Thank you for your guidance. CpNAC1 was screened using the wintersweet transcriptome database(Liu et al., 2014). We will conduct further research about the NAC TF and their biological relevance in Chimonanthus praecox in our next experiment.

Point 2: Figure legends less descriptive specially Figure 2 legend. In General Figure legends need to be explained more to clear the message, specially figure 2 part B, BP and VP are not defined….

Response 2: Thank you for your guidance. We have added some message in Figure 2 legend in line 145, to explain the pGBKT7-VP and pGBKT7, as “ Transactivation activity analysis of CpNAC1. The plasmid of pGBKT7-CpNAC1, pGBKT7-VP (positive control), and pGBKT7 (negative control) were transformed into the AH109 yeast strain. ”.

Point 3: Expression of CpSNAC gene could be first presented and discussion about other isoforms as transcriptomic data available and then follow by isolation and characterization….

Response 3: Thank you for your guidance. CpNAC1 was screened using the wintersweet transcriptome database(Liu et al., 2014). Other NAC family genes in wintersweet will be studied in subsequent experiments.

Point 4: It is not clear why author name CpSNAC instead of CpNAC… S trends for what!!

Response 4: Thank you for your guidance, we have changed the name as “CpNAC1”. 

Point 5: Do the author try to prepare transgenic line in endogenous species (C. praecox)? What is the biological relevance to study CpSNAC…. Authors claim it would be helpful to know the mechanism of abiotic stress tolerance that is far from what author presented in this study.

Response 5: Thank you for your guidance. The preparation of transgenic lines in endogenous species is the most important research work in our laboratory at present, and we also expect to establish the genetic transformation system of Chimonanthus praecox in the future. we referred to the wintersweet transcriptome database and previous research work in our laboratory(Liu et al., 2014; Lin et al., 2021), we suspected that CpNAC1 is involved in abiotic stress tolerance. We isolated CpNAC1 promoter and gene from wintersweet, then analysis the expression pattern under abscisic acid(ABA) induction and abiotic stress(42℃、4℃、PEG、NaCl) treatment . We found that CpNAC1 promoter and gene can be induced by NaCl、PEG and ABA, we hypothesized that the response of this gene to abiotic stress is related to the ABA pathway. So we're going to continue to work on that.

Point 6: Phenotypic experiments are not sufficient to conclude that CpNAC negatively regulates the osmotic tolerance… more experiment can be done i.e., root growth assay in vitro control conditions and stress conditions… the data presented is not clear enough to see the phenotype… what the rationale between choosing stress conditions for example concentration of ABA and Nacl…

Response 6: Thank you for your guidance. In this study, we focused more on the promoter and studied the induced expression of CpNAC1 from the transcriptional level. In the future, we will conduct more experiments on transgenic Arabidopsis thaliana. The concentration of ABA and Nacl, we referred to the previous research work in our laboratory(Tian et al., 2020; Hu et al. 2021).

Point 7: What will happen if the osmotic stress prolonged and is it possible to recover the phenotype?

Response 7: Thank you for your guidance. If the osmotic stress prolonged and it is possible to recover the phenotype, and it is negatively correlated with the concentration of the PEG solution and the treatment time(Zhang et al.,2012).

Point 8: How yeast cells were transformed .. method is not described and also lack of information on the method used to study the transcriptional activity assay in yeast.

Response 8: Thank you for your guidance. we have added it in line 392-395, as “ according to the manufacturer’s instructions (Clontech); this was performed according to the yeast protocol handbook (Clontech)”. 

Point 9: Which method used for RT-qPCR analysis. The results of qPCR is not well discussed in discussion that deserved to be discussed more clearly in discussion.

Response 9: Thank you for your guidance, we have added it in line 300, 307 and 321, as “ The results suggest that CpNAC1 may function in flower organ development and abiotic stress in early plant growth.;  This may be related to their functional diversity; These results are inconsistent with the specific expression results of CpNAC1 in the tissues of wintersweet. CpNAC1 is derived from the genome of wintersweet. There are differences in organ structure between woody plants and herbaceous plants, and the upstream regulatory sequences of the two plants are different, so the efficiency and location of the promoter may also be different”. 

Point 10:  What kind of statistical test used to test the significant level that need to be mentioned in figure legends or in methodology.

Response 10: Thank you for your guidance, we have added methods of difference significance analysis of each result in materials and methods in line 461.

Point 11:  Finally I would suggest a slight modification of  the title: Functional characterization of the CpNAC1 promoter and gene from Chimonanthus praecox in Arabidopsis

Response 11: Thank you for your guidance, we have modified the title, as “Functional characterization of the CpNAC1 promoter and gene from Chimonanthus praecox in Arabidopsis”.

Reviewer 3 Report

Zhao et al. present a functional characterization of the Chimonanthus praecox NAC1 (CpSNAC1) transcription factor (TF) through gene expression analysis, promoter activity, subcellular localization of CpSNAC1 in tobacco leaves and tissue-specific activity in transgenic Arabidopsis lines.

The authors have made an effort to identify and analyze the role of TF CpSNAC1, which is unpublished information in the scientific literature. The story is interesting, and it should be considered by International Journal of Molecular Sciences. However, there are some minor mistakes in the manuscript, that the authors should check it carefully before this manuscript is ready for publication.

1 - Within the large family of NAC TF, for what reasons did the authors choose to specifically study CpSNAC1?

2 – Through qRT-PCR analyses CpSNAC1 expression was highest in mature leaves and flower-bud stage, but GUS staining show activity in all tissues tested in Arabidopsis transgenic lines. That is, the GUS data did not corroborate the RT-qPCR data. Thus authors should discuss that differences in tissue-specific activity pattern.

3- Which statistical test was used in data shown in Figure 3, 5, 6, 7 and 8?

4 - Lines overexpressing CpSNAC1 should be better described. Which promoter was chosen to control CpSNAC1? What is the level of heterologous CpSNAC1 expression in each of the three Arabidopsis transgenic lines?

5 -  I think that a specific conclusion about the CpSNAC1 gene, that was actually studied in this work and not about the entire CpSNAC family would be more appropriate.

Line 24 - Consider change to " had high GUS...

Line 113 - Indicate in parentheses to which species each of these homologous genes belongs.

Author Response

Response to Reviewer 3 Comments

Point 1: Within the large family of NAC TF, for what reasons did the authors choose to specifically study CpSNAC1?

Response 1: Thank you for your guidance. CpNAC1, a NAC family gene, was screened using the wintersweet transcriptome database(Liu et al., 2014). We will conduct further research about the NAC TF and their biological relevance in Chimonanthus praecox in our next experiment.

Point 2: Through qRT-PCR analyses CpSNAC1 expression was highest in mature leaves and flower-bud stage, but GUS staining show activity in all tissues tested in Arabidopsis transgenic lines. That is, the GUS data did not corroborate the RT-qPCR data. Thus authors should discuss that differences in tissue-specific activity pattern.

Response 2: Thank you for your guidance. we have added the discussion in line 321, as “These results are inconsistent with the specific expression results of CpNAC1 in the tissues of wintersweet. CpNAC1 is derived from the genome of wintersweet. There are differences in organ structure between woody plants and herbaceous plants, and the upstream regulatory sequences of the two plants are different, so the efficiency and location of the promoter may also be different”.

Point 3: Which statistical test was used in data shown in Figure 3, 5, 6, 7 and 8?

Response 3: Thank you for your guidance, we have added methods of difference significance analysis of each result in materials and methods in line 461.

Point 4:  Lines overexpressing CpSNAC1 should be better described. Which promoter was chosen to control CpSNAC1? What is the level of heterologous CpSNAC1 expression in each of the three Arabidopsis transgenic lines?

Response 4: Thank you for your guidance, we have added some message to describe the lines overexpressing CpNAC1 in line 446, as “The plant overexpression vector was constructed by inserting CpNAC1 into the pCAMBIA1300 vector (containing the CaMV35S promoter) using specific primers pCAMBIA1300-CpNAC1-F/R (Table S1) with SacI and Xbal sites, then transformed into Agrobacterium tumefaciens strain GV3101 via electroporation. Arabidopsis was transformed as described in subsection 4.5 ”. 

Point 5: I think that a specific conclusion about the CpSNAC1 gene, that was actually studied in this work and not about the entire CpSNAC family would be more appropriate.

Response 5: Thank you for your guidance. We have modified this statement in line 29 and 343.

Point 6: Consider change to " had high GUS...

Response 6: Thank you for your guidance. We have changed it in line 22, as “showed high GUS...”.

Point 7:  Indicate in parentheses to which species each of these homologous genes belongs.

Response 7: Thank you for your guidance. we have added species of these homologous genes in line 105.

Round 2

Reviewer 2 Report

Comments on R1:

Authors tried to address some of my points however for complete story, there are still some gaps. Here I have mentioned a few more points that could be considered to improve the MS. After thease modifications, this MS can be published in IJMS.

Point 1: Lack of background/rationale behind the choice of the gene of interest. There is no discussion how many homologs of NAC TF are present in Chimonanthus praecox and their biological relevance in C. praecox.

Response 1: Thank you for your guidance. CpNAC1 was screened using the wintersweet transcriptome database(Liu et al., 2014). We will conduct further research about the NAC TF and their biological relevance in Chimonanthus praecox in our next experiment.

Point1.1: Include a justification of the isoform in introduction.

Point 3: Expression of CpSNAC gene could be first presented and discussion about other isoforms as transcriptomic data available and then follow by isolation and characterization….

Response 3: Thank you for your guidance. CpNAC1 was screened using the wintersweet transcriptome database(Liu et al., 2014). Other NAC family genes in wintersweet will be studied in subsequent experiments.

Point 3.1: that’s ok. But at least try to mentioned how many homologs they possess and existing literature in introduction for a proper rationale of your project.

Point 5: Do the author try to prepare transgenic line in endogenous species (C. praecox)? What is the biological relevance to study CpSNAC…. Authors claim it would be helpful to know the mechanism of abiotic stress tolerance that is far from what author presented in this study.

Response 5: Thank you for your guidance. The preparation of transgenic lines in endogenous species is the most important research work in our laboratory at present, and we also expect to establish the genetic transformation system of Chimonanthus praecox in the future. we referred to the wintersweet transcriptome database and previous research work in our laboratory(Liu et al., 2014; Lin et al., 2021), we suspected that CpNAC1 is involved in abiotic stress tolerance. We isolated CpNAC1 promoter and gene from wintersweet, then analysis the expression pattern under abscisic acid(ABA) induction and abiotic stress(42℃4℃PEGNaCl) treatment. We found that CpNAC1 promoter and gene can be induced by NaClPEG and ABA, we hypothesized that the response of this gene to abiotic stress is related to the ABA pathway. So we're going to continue to work on that.

Point 5.1: Please include this point in discussion…. [ we suspected that CpNAC1 is involved in abiotic stress tolerance. We isolated CpNAC1 promoter and gene from wintersweet, then analysis the expression pattern under abscisic acid(ABA) induction and abiotic stress(424PEGNaCl) treatment. We found that CpNAC1 promoter and gene can be induced by NaClPEG and ABA, we hypothesized that the response of this gene to abiotic stress is related to the ABA pathway.]

Point 6: Phenotypic experiments are not sufficient to conclude that CpNAC negatively regulates the osmotic tolerance… more experiment can be done i.e., root growth assay in vitro control conditions and stress conditions… the data presented is not clear enough to see the phenotype… what the rationale between choosing stress conditions for example concentration of ABA and Nacl…

Response 6: Thank you for your guidance. In this study, we focused more on the promoter and studied the induced expression of CpNAC1 from the transcriptional level. In the future, we will conduct more experiments on transgenic Arabidopsis thaliana. The concentration of ABA and Nacl, we referred to the previous research work in our laboratory(Tian et al., 2020; Hu et al. 2021).

Point 6.1: That need to be clearly referred in manuscript. It can be referred in method line number 401-402.

Point 7: What will happen if the osmotic stress prolonged and is it possible to recover the phenotype?

Response 7: Thank you for your guidance. If the osmotic stress prolonged and it is possible to recover the phenotype, and it is negatively correlated with the concentration of the PEG solution and the treatment time(Zhang et al.,2012).

Point 7.1: Include this information in discussion with reference.

Point 8: How yeast cells were transformed. method is not described and also lack of information on the method used to study the transcriptional activity assay in yeast.

Response 8: Thank you for your guidance. we have added it in line 392-395, as “according to the manufacturer’s instructions (Clontech); this was performed according to the yeast protocol handbook (Clontech)”.

Point 8.1: Please specify the exact yeast transformation protocol… I guess it is LiAc Yeast Transformation Procedure.

Author Response

Response to Reviewer 2 Comments

Point 1: Lack of background/rationale behind the choice of the gene of interest. There is no discussion how many  homologs of NAC TF are present in Chimonanthus praecox and their biological relevance in C. praecox

Response 1: Thank you for your guidance. CpNAC1 was screened using the wintersweet transcriptome database(Liu et al., 2014). We will conduct further research about the NAC TF and their biological relevance in Chimonanthus praecox in future experimental studies.

Point1.1: Include a justification of the isoform in introduction.

Response 1.1CpNAC1 was screened was based on the second generation transcriptome database constructed previously in our laboratory(Liu et al., 2014). Then we analysised the expression pattern of CpNAC1 under different abiotic stress, and found that this gene could respond to a variety of stresses, including 42℃、4℃、PEG、NaCl treatment. Hence, we cloned this gene and its promoter sequence from wintersweet and we more focuses on the induction of promoters.

Point 3: Expression of CpSNAC gene could be first presented and discussion about other isoforms as transcriptomic data available and then follow by isolation and characterization….

Response 3: Thank you for your guidance. CpNAC1 was screened using the wintersweet transcriptome database(Liu et al., 2014). Other NAC family genes in wintersweet will be studied in subsequent experiments.

Point 3.1: that’s ok. But at least try to mentioned how many homologs they possess and existing literature in introduction for a proper rationale of your project.

Response 3.1De novo assembly of the wintersweet transcriptome in the absence of a reference genome using Trinity De novo transcriptome assembly softwar. Our transcriptome database obtained 106,995 transcripts which may have redundancy and assemble errors. It is difficult for us to identify the whole homologous gene of NAC in wintersweet.

Point 5: Do the author try to prepare transgenic line in endogenous species (C. praecox)? What is the biological relevance to study CpSNAC…. Authors claim it would be helpful to know the mechanism of abiotic stress tolerance that is far from what author presented in this study.

Response 5: Thank you for your guidance. The preparation of transgenic lines in endogenous species is the most important research work in our laboratory at present, and we also expect to establish the genetic transformation system of Chimonanthus praecox in the future. we referred to the wintersweet transcriptome database and previous research work in our laboratory(Liu et al., 2014; Lin et al., 2021), we suspected that CpNAC1 is involved in abiotic stress tolerance. We isolated CpNAC1 promoter and gene from wintersweet, then conducted abscisic acid(ABA) induction and abiotic stress(42℃、4℃、PEG、NaCl) expression pattern. We found that CpNAC1 promoter and gene can be induced by NaCl、PEG and ABA, we hypothesized that the response of this gene to abiotic stress is related to the ABA pathway. So we're going to continue to work on that.

Point 5.1: Please include this point in discussion…. [ we suspected that CpNAC1 is involved in abiotic stress tolerance. We isolated CpNAC1 promoter and gene from wintersweet, then analysis the expression pattern under abscisic acid(ABA) induction and abiotic stress(42℃、4℃、PEG、NaCl) treatment. We found that CpNAC1 promoter and gene can be induced by NaCl、PEG and ABA, we hypothesized that the response of this gene to abiotic stress is related to the ABA pathway.]

Response 5.1:Thank you for your guidance. According to the original manuscript, we have adjusted and added this point in discussion in line 311-317.

Point 6: Phenotypic experiments are not sufficient to conclude that CpNAC negatively regulates the osmotic tolerance… more experiment can be done i.e., root growth assay in vitro control conditions and stress conditions… the data presented is not clear enough to see the phenotype… what the rationale between choosing stress conditions for example concentration of ABA and Nacl…

Response 6: Thank you for your guidance. In this study, we focused more on the promoter and studied the induced expression of CpNAC1 from the transcriptional level. In the future, we will conduct more experiments on transgenic Arabidopsis thaliana. The concentration of ABA and Nacl, we referred to the previous research work in our laboratory(Tian et al., 2020; Hu et al. 2021).

Point 6.1: That need to be clearly referred in manuscript. It can be referred in method line number 401-402.

Response 6.1:Thank you for your guidance. We have added the relevant references in line 401 and 404.

Point 7: What will happen if the osmotic stress prolonged and is it possible to recover the phenotype?

Response 7: Thank you for your guidance. If the osmotic stress prolonged and it is possible to recover the phenotype, and it is negatively correlated with the concentration of the PEG solution and the treatment time(Zhang et al.,2018).

Point 7.1: Include this information in discussion with reference.

Response 7.1:Thank you for your guidance. We have added this information in discussion in line 341.

Point 8: How yeast cells were transformed .. method is not described and also lack of information on the method used to study the transcriptional activity assay in yeast.

Response 8: Thank you for your guidance. we have added it in line 392-395, as “ according to the manufacturer’s instructions (Clontech); this was performed according to the yeast protocol handbook (Clontech)”. 

Point 8.1: Please specify the exact yeast transformation protocol… I guess it is LiAc Yeast Transformation Procedure.

Response 8.1:Thank you for your guidance. That's right, it's the LiAc-mediated transformation method. we have added it in line 394.
